# Bio-Impedance Sensor for Real-Time Artery Diameter Waveform Assessment

**DOI:** 10.3390/s21248438

**Published:** 2021-12-17

**Authors:** Mugeb Al-harosh, Marat Yangirov, Dmitry Kolesnikov, Sergey Shchukin

**Affiliations:** Department of Medical and Technical Information Technology, Bauman Moscow State Technical University, 105005 Moscow, Russia; yangirovma@student.bmstu.ru (M.Y.); Kolesnikovda1@student.bmstu.ru (D.K.); schookin@bmstu.ru (S.S.)

**Keywords:** artery diameter waveform, red cell orientation effect, blood pressure, bio-impedance

## Abstract

The real-time artery diameter waveform assessment during cardio cycle can allow the measurement of beat-to-beat pressure change and the long-term blood pressure monitoring. The aim of this study is to develop a self-calibrated bio-impedance-based sensor, which can provide regular measurement of the blood-pressure-dependence time variable parameters such as the artery diameter waveform and the elasticity. This paper proposes an algorithm based on analytical models which need prior geometrical and physiological patient parameters for more appropriate electrode system selection and hence location to provide accurate blood pressure measurement. As a result of this study, the red cell orientation effect contribution was estimated and removed from the bio-impedance signal obtained from the artery to keep monitoring the diameter waveform correspondence to the change of blood pressure.

## 1. Introduction

Arterial blood pressure is the main physiological indicator of cardiovascular system [1,2]. However, a deviation from normal range can be used to identify different diseases such as thrombosis and atherosclerosis, which can lead to mortality if the medical intervention is not started on time [3,4]. The arterial blood pressure parameters tend to change over time, so long-term monitoring helps to provide more accurate evaluation of the patient’s condition, as well as beat-by-beat measurement of blood pressure, which could offer several advantages toward the early detection and the patient response to specific therapy and medication [5]. Currently, there are several approaches for blood pressure measurement [6,7,8,9,10,11,12,13,14]. The blood pressure assessment by sphygomomanometry is the widely used method in medical practice due to its simplicity and sufficient accuracy; however, this method cannot provide long-term measurements due to frequent occlusion [11]. The noninvasive arterial blood pressure waveform estimation based on an arterial cross-sectional area measurement combined with an elasticity measurement of the vessel has been developed using ultrasound [6]. However, the devices created by scientists based on these approaches are either stationary or more dimensional. The optical based methods are unreliable for obese patients, while the tonometry methods require a specific anatomical structure with superficial artery supported by bone and so is sensitive to the device placement [15,16,17]. The analysis of blood pressure estimation methods is summarized in Table 1.

Recently, the pulse transit time-based methods have been widely used in order to develop a wearable device that can measure blood pressure continuously. Two sensors located at two different sites can measure the pulse waveforms, which can be used to calculate the pulse transit time (PTT) value. The change of the PTT value shows a strong correlation with blood pressure [3]; however, the standard model equation used for this calculation depends on the artery diameter, which cannot be identified for large measurement segments [35]. In [6], there is an attempt to develop a portable blood pressure device based on local PTT measurements. However, the mathematical model used in this work is identified for the volume change in the whole segment under study, which makes the estimation based on this model not accurate and not related to diameter change. Thus, the aim of this study is to develop a wearable bio-impedance sensor for blood pressure waveform monitoring. This method is based on electrical impedance measurement in a small segment with uniform artery diameter and depth. The sensor provides regular measurement of the pressure-dependent physiological parameters such as the artery diameter waveform and elasticity, which allow the accurate measurement of blood pressure. This work proposes the use of a mathematical model for diameter waveform assessment; however, the required ultrasound-based information before the measurement provides specific sensor parameters, which lead to a more accurate determination of blood pressure. According to [35,36,37], the arterial wall elasticity changes with the blood pressure change; thus, this work proposes an additional measurement of pulse transit time, which can help the regular estimation of arterial elasticity change. The idea of red cell orientation effect estimation was suggested by [38,39] to better understand the impedance cardiography waveforms. However, the idea of understanding the orientation effect contribution to the peripheral bio-impedance signal obtained from the specific small segment illustrated in this study is the first according to our knowledge.

As a result, a key objective of this study is to improve the accuracy of the existing algorithm based on PTT measurement to estimate the arterial blood pressure by taking into account the contribution of red cell orientation effects as well as the artery diameter and stiffness change during long-term monitoring.

## 2. Materials and Methods

### 2.1. Mathematical Model

For artery diameter waveform assessment from the corresponding electrical bio-impedance signal, it is required to develop a justified mathematical model which characterizes the region of interest with full details [40]. To solve this problem, both analytical and numerical methods can be used. As a rule, numerical methods for solving problems, describing the electric current distribution through biological tissues, involve powerful computing systems, which do not solve the issue of calculating the necessary numerical estimates in real time, inherent in wearable monitoring systems. However, in the case of a semi-homogenous study area, the analytical approaches with some assumptions are promising towards the solution of this problem. Accordingly, the region of interest should be a place with a peripheral artery that can be isolated from the adjacent artery branches and veins to obtain the required bio-impedance measurement without any deterioration and additional sources that can affect the desired signal from the selected artery. However, the region of interest, where the major arteries widely used for such measurement [41] are the carotid, brachial, and radial arteries, comprises such tissues which have different conductivity [42]. The electrical resistivities of different tissues in the region of interest as a function of frequency are shown in Figure 1.

Thus, an electrical current with 100 kHz was considered in this work as the impact of the capacitive part, which was determined by the heterogeneity of tissue structures in the region of interest as less than 10%. Morever, a four-electrode system configuration, which is shown in Figure 2, was proposed in this study to provide the desired accuracy for bio-impedance measurement; thus, two outer electrodes were used to inject a small alternating current into the human body. The voltage drop between the inner electrodes was measured to calculate the impedance, while the electrode impedances did not contribute to the measured voltage.

Thereby, for small segment selection, where the change of blood volume is dominated by the artery diameter, it is acceptable to join all tissues around the artery with common apparent resistivity $\rho_{1\;}$. This assumption could minimize the requirement of the mathematical model as well as the calculation time without losing the desired accuracy. As a result, an analytical solution for the mathematical model of uniform space with cylinder inclusion was proposed in this work, while the artery and the surrounding tissues are modeled by a cylinder and uniform half-space, respectively, as shown in Figure 3.

In the proposed mathematical model, the electrical potential is a solution of Laplace’s equation in cylindrical coordinates with appropriate boundary conditions associated with a cylinder [44]. This model considers orthogonal electrode system placement regarding the artery, and with prior information about the artery depth and diameter, the appropriate electrode system can be set. The analytical solution of the proposed mathematical model is a function, which depends on serval parameters shown in Equation (1). The limit values for the model parameters that can be measured are illustrated in Figure 3. Thus, depending on the prior obtained information, the doctor can also select the right place for sensor placement, which can be the brachial, radial, or carotid artery depending on patient obesity.
(1)$$Z=func\left(a,b,r,h,x,\rho_{1},\rho_{2},\right)$$
where *a* is the distance between current electrodes, *b* the distance between potential electrodes, *r* the artery radius, *h* artery depth, *x* is the displacement of the electrode system relative to the center of the artery. This work proposes accurate placement of the electrode system regarding the artery position, which is why the x parameter is equal to zero.

The apparent resistivity $\rho_{1}$ of the surrounding tissues can be calculated from the measured base impedance value according to [40] which is a quasi-constant value, while the resistivity of blood $\rho_{2}$ is a function, which depends on different factors such as hematocrit, temperature, viscosity, velocity, and share rate [45,46].

Basically, for artery diameter change estimation based on the corresponding bio-impedance measurement, the inverse problem for the proposed model should be obtained as shown in Equation (2). However, this approach requires multichannel measurements and sophisticated calculation, which make it undesirable for a portable monitoring system.
(2)$$dR=func\left(a,b,r,h,x,\rho_{1}+\Delta \rho_{1},\rho_{2},dZ\right)$$

The initial calibration step which requires prior information such as the artery diameter and depth as well as the blood velocity allows minimizing the multi-parameter model to one unknown parameter function. The only unknown dynamic parameter of the proposed model is the diameter change of a cylinder, which simulates the artery diameter change and the blood volume change in arteries. This unknown parameter can be estimated from the measured pulse impedance from an artery by means of the dependence dR = func(dZ), which is the inverse of the forward dependence dZ = func(dR) that can be obtained on the basis of the proposed analytical model. Figure 4 shows the pulse impedance change depending on artery diameter change and vice versa. The observed decrease in impedance with an increase in artery diameter can be explained by the fact that blood is a more conductive medium comparing with the surrounding tissues. However, since the function dZ = func(dR) is monotone, then there is a unique inverse solution for this function. Thus, an iterative algorithm to address the linear inverse value of the diameter change was proposed in this paper.

This model, nevertheless, has an assumption of point electrodes and does not take into consideration the electrodes’ dimensions. It was verified for adequacy in previous study by this author [40] and concluded for its application to resolve the forward and inverse problem. Moreover, the use of spherical electrodes can minimize this error as the current distribution from a point physically not differing from the sphere.

### 2.2. Red Blood Cell Orientation Effect Estimation

The generation of shear forces across the width of a blood artery vessel during flow causes the cells to align with the minimal cross-sectional area facing the direction of flow. This in turn results in a larger cross-sectional area of plasma and a reduction in the resistivity of the blood as the flow increases [38]. Understanding the contribution of this effect on the peripheral impedance signal is a vital step in achieving the artery diameter waveform [12,13,36]. The experimental study, which was conducted in vitro on rigid tubes [38], showed a blood resistivity change of 15–30% due to erythrocyte orientation, and according to [38,39], the change of blood resistivity due to velocity can be estimated using Equation (3). This equation was deduced for the longitudinal electrode system position regarding the artery axis. However, the red cell contribution of the blood electrical resistivity change on the orthogonal direction is the same in value but with opposite direction [47].
(3)$$\Delta \rho_{2}=-0.45\cdot H\cdot \rho_{2}\cdot \left(1-exp\left(-0.26\cdot {\left(\bar{\gamma }\right)}^{0.39}\right)\right),$$
where $\rho_{2}$ is the blood electrical resistivity, $\bar{\gamma }$ is the shear rate averaged over the cross section of the vessel, and *H* is the hematocrit [48].

The shear rate averaged over the cross-section of the vessel is determined by Equation (4) [49].
(4)$$\bar{\gamma }=\frac{2\cdot V_{max}}{R}\cdot \left(\frac{n}{n+1}\right),$$
where $V_{max}$ is the maximum velocity in the center of the vessel estimated using spectral Doppler [50,51], *R* is the radius of the vessel, *n* is the exponent of the power law, *n* = 2 for a parabolic profile and more than 2 for a flat profile.

### 2.3. Elastic Modulus Estimation

The elastic response of a blood vessel can be expressed in terms of compliance, dispensability, stiffness, or elastic modulus [52,53]. According to [53], the elasticity modulus is not a constant parameter and can vary in a wide range, depending on internal and external factors. These changes will cause an additional error in blood pressure; thus, the blood pressure estimation is based on mathematical models, which contain such parameters as the diameter change and the elasticity modulus. The elastic modulus is related to the pulse wave velocity and the artery thickness by the Moens–Korteweg Equation (5) below [54].
(5)$$E=\frac{V^{2}\cdot \rho \cdot d}{h_{a}},$$
where *V* is the pulse wave propagation speed, *E* is elastic modulus of the artery wall, ρ is the arterial blood density, *d* is the radius of the artery, *h_a_* is the vessel thickness.

The measurement of local pulse wave velocity (PWV), which can be achieved by measuring the pulse waveform from two different points, can determine the actual elasticity of the artery in the region of interest, while adding a sensor that can provide the monitoring of this value and recalibrate the system according to this change can allow the accurate beat-to-beat blood pressure measurement.

### 2.4. Experimental Setup

The experimental study was conducted at the medical center of Bauman Moscow State Technical University and the research ethics was followed. Four healthy young subjects were involved in this study. The measurements were performed on the arm in the area of the elbow joint, at the distal end of the brachial artery before the bifurcation into radial and ulnar arteries. The advantage of this location is the access to control the right electrode position regarding the artery location [55]. The General Electric model LOGIC S8 with an ML6-15-D ultrasonic transducer at a frequency of 6.3 MHz was used during this study to obtain prior information such as the arterial blood velocity, diameter, depth, and the wall thickness as shown in Figure 5. Figure 5a shows the brachial artery velocity, which can be used for red cell orientation effect estimation, while the artery diameter and depth were also determined. These data have been used as inputs for the mathematical model to select the appropriate electrodes system as well as the further estimation of red cell orientation effect and the brachial artery waveform diameter. The blood velocity curves for all subjects obtained using US were digitized and averaged as shown in Figure 5b.

The multichannel system Reo-32 was used for bio-impedance measurement. The system provides a 4-electrode technique by applying an alternating current of frequency of 100 kHz and constant amplitude to a 3-mA-current electrode, and provides simultaneous measurement through different measuring channels [56]. The technical specifications of the multichannel system are illustrated in Table 2.

The electrode system array shown in Figure 6 was designed for the multichannel measurement. The electrode array provides electrical impedance mapping from 5 adjacent points. Dry electrodes with a diameter of 5 mm and a distance between adjacent electrodes of 7 mm were used to provide symmetrical and identical requirements for measurements for the bio-impedance signal obtained from the artery, which depend also on the electrode system spacing. The advantage of this electrode system is the ability to control the desired artery position.

Figure 6a shows the experimental setup and the electrode placement in the region of interest. Skin preparation prior to electrode placement on a patient was performed to increase the conductivity for the dry electrodes. The right location of the electrode system was controlled by the obtained signals from all the measuring channels. Non-inversion behavior of signals was observed, which testifies that the whole artery length was located in the positive sensitivity region of the electrode system.

The obtained bio-impedance signals were used for both diameter estimation and the local brachial artery stiffness index, which were performed from 2 channels located in parallel at a distance of L = 21 mm as shown in Figure 6b.

## 3. Results

As shown in Figure 7a, the blood velocity curve obtained from US was digitized and approximated by linear interpolation, then filtered by a Savitzky–Golay filter to get an adequate signal for shear rate estimation. Thus, Savitzky–Golay filtering can provide accurate smoothing by using polynomial approximation of local least-squares [57]. The obtained filtered curve of blood velocity was used for share rate assessment using Equation (4), while the exponent of the power law (n) was equal to 2 as the velocity profile was parabolic. Figure 7b shows the obtained share rate curve, which has been used as an input for the blood cell orientation effect estimation model shown in Equation (3). The hematocrit value during this calculation was accepted to be equal to 47%, which corresponds to a normal healthy value.

Figure 8 shows the blood resistivity change due to red cell orientation effect for one of the involved subjects, which corresponds to a 16% increase compared with the initial blood resistivity value 1.35 Ohm·m. [58]. The obtained results from the first step showed the high contribution of brachial blood velocity in the electrical impedance change, respectively; the change behavior of this value during the cardiac cycle should be taken into consideration for the following steps and calculations.

Figure 9 shows the electrical bio-impedance signals obtained from the brachial artery. Simultaneously, the base impedance and the pulse impedance were recorded. Nevertheless, the designed electrode system provides simultaneous measurement from five adjacent locations; two signals were selected for further processing. The selected signals were far from each other by 21 mm, which were enough to get the PTT [6,58]. Moreover, this distance was enough to select a uniform segment of artery as the recoding signals were equal in amplitude but have a shift in time due to different location of the measuring electrodes.

The analytical solution of the proposed mathematical model is an infinite sum of Bessel functions. Thus, in order to reduce the computation time, an algorithm based on the first and the second derivative was developed for five points per cardiac cycle extraction from the measured pulse waveforms. The first point is the diastolic point that corresponds to the maximum of the second derivative; the second point is the maximum slope point in the middle of the ascending point section that corresponds to the maximum of first derivative; the third point is the systolic point that corresponds to the zero value of the first derivative; the fourth point is the dicrotic notch that corresponds to the maximum of the second derivative; and the fifth point is the reflection point. The selected points are necessary for full restoration of the diameter waveform shape and its correct amplitude; thus, the time derivative of the bio-impedance signal was obtained by the following transfer function:(6)$$H\left(Z\right)=\frac{1}{10T}\left(-2Z^{-2}-Z^{-1}+Z^{1}+2Z^{2}\right)$$
where *T* is the sampling period; *z* is the recorded bio-impedance signal.

According to the obtained data from the previous sections, the diameter waveform was estimated by the following algorithm: as the electrode system was located against the artery, the red cell orientation effect increased the blood electrical resistivity; correspondingly, the ∇ρ shown in Figure 8 was added to the actual blood resistivity $\rho_{1}$, which was equal to 1.35 Ohm.m. The rest of the model parameters such as the radius and depth are constant as shown in Equation (7).
(7)$$\Delta Z_{m\;}=func\left(a,b,r,h,x,\rho_{1},\rho_{2}+\Delta \rho_{2},dR\right)$$

The base impedance was used to estimate the apparent resistivity $\;\rho_{1}$ of the region of interest according to Equation (8) [40], while the $\Delta Z_{m}$ is the recorded pulse impedance signal shown in Figure 9.
(8)$$Zbase=\frac{2\rho_{1}b}{\pi \left(a^{2}-b^{2}\right)}\;$$

For every selected point based on the derivative-based algorithm, the artery diameter change was evaluated and, correspondingly, the diameter artery waveform was restored. Figure 10 shows the obtained diameter change for the four involved subjects; these curves show the percentage change in artery diameter relative to diastolic diameter **D_D_**. However, the obtained curves of diameter waveform reflect the nature of blood pressure change during the cardio cycle.

Basically, the local brachial elasticity was estimated from the time delay between the two recorded signals. The maximum of the first derivative was selected as a specific point to get the PTT between two points with an inaccuracy of ±0.001 s corresponding to a sample frequency of 500 Hz. According to Equation (5) and based on the obtained PTT values, the local brachial artery elastic modulus was determined and illustrated in Table 3: the obtained values correspond to healthy arteries [59]. However, these values can vary significantly with blood pressure change and the bio sensor should be recalibrated according to the possible change.

## 4. Discussion

This work presents a new method for real-time arterial diameter waveform estimation, which can improve the accuracy of blood pressure measurement. This method is based on an electrical bio-impedance signal obtained from an artery using appropriate electrode system with dry electrodes. The bio-impedance pulse obtained from the artery dominantly contains different mechanisms, which have absolutely different behaviors depending on the location of the bio-impedance sensor against the artery. According to the proposed electrode system position in this study, the diameter change leads to increase in the blood conductivity while the velocity effect goes in the absolute opposite direction, which makes the diameter waveform assessment a big challenge to be extracted from the integral signal that has these two coherent mechanisms. In this study, a mathematical model, which is a function of different geometrical and electrophysiological parameters, was proposed to get the artery diameter waveform. However, the inverse problem of this model requires sophisticated mathematical calculations, which makes it not suitable for a wearable device for the blood pressure monitoring. Alternatively, the prior information required for this study using the US allowed minimizing the multi-parameter model to one unknown parameter function. The change of the peripheral arteries’ diameter during the cardio cycle was very low compared with its own diameter, hence, the electrical bio-impedance change due to this variation was also very low. Thus, the prior calculation using the mathematical model helps to select not only the appropriate electrode system parameters but also the right arterial site position for measuring the bio-impedance signal, which can be in the brachial, carotid, or radial artery, in order to provide accurate blood pressure measurement. The effect of the red cell orientation on the electrical bio-impedance of pulsatile blood flow through the brachial artery, which was considered in this study, is the first for the implications of artery diameter waveform estimation. Although the pulse transit time has been a promising non-invasive and cuff-less method to measure blood pressure, the blood pressure–PTT relationship, which is modelled by the Moens–Korteweg, was derived from a simplified mechanical model, which was insensitive to the artery diameter change. This can lead to an error with arterial pressure measurement, especially in young patients where the artery diameter can be noticeably changed with the change of blood pressure. Thus, improving the accuracy of the existing algorithm based on PTT measurement to estimate the arterial blood pressure, by taking into account the contribution of red cell orientation effects and the artery diameter and stiffness change during long-term monitoring, is a very important task and this paper completely covers and answers all of these lacking questions.

## Figures and Tables

**Figure 1 sensors-21-08438-f001:**
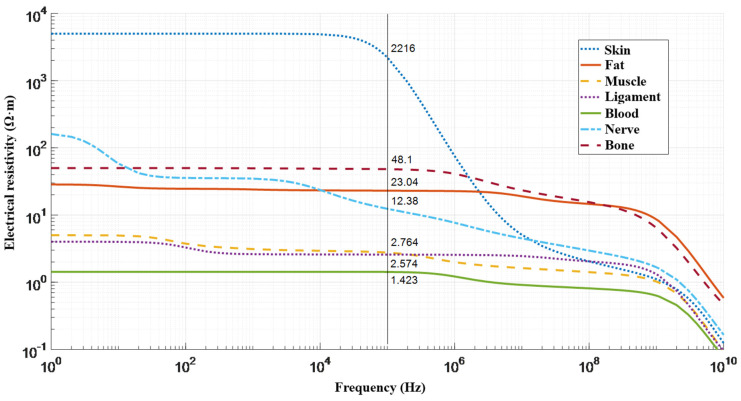
Resistivities for various human tissues in the region of interest as a function of frequency [43].

**Figure 2 sensors-21-08438-f002:**
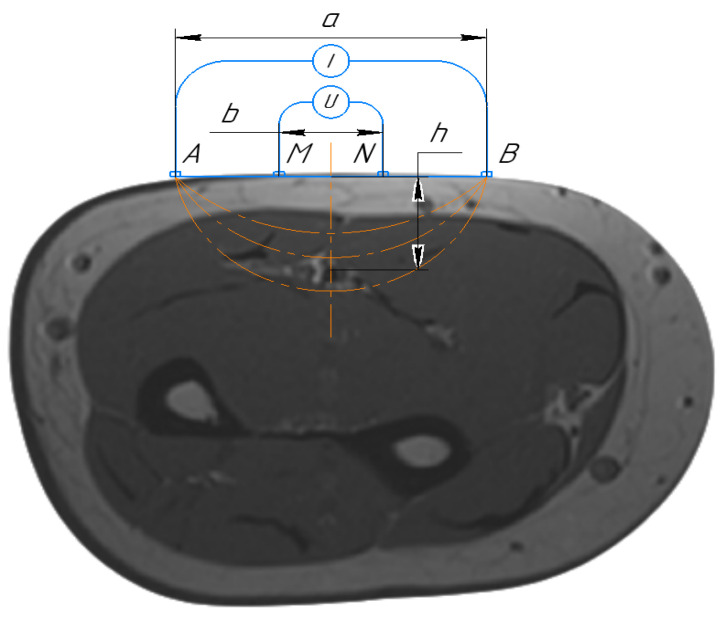
Schematic diagram of the four-electrode setup. A and B are the current electrodes, M and N are the measuring electrodes.

**Figure 3 sensors-21-08438-f003:**
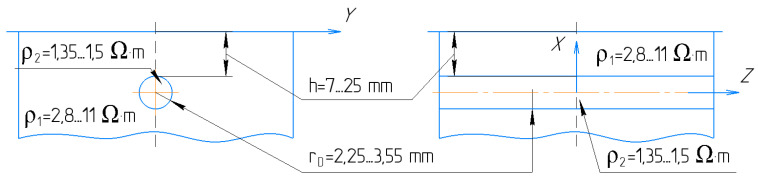
The mathematical model of artery in the region of interest.

**Figure 4 sensors-21-08438-f004:**
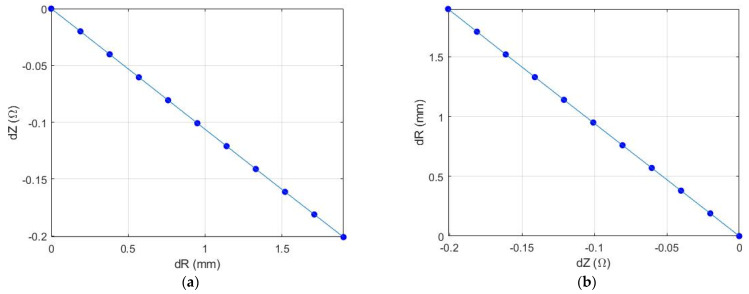
The dependence of electrical impedance changes due to artery diameter change: (**a**) The forward problem; (**b**) the inverse problem.

**Figure 5 sensors-21-08438-f005:**
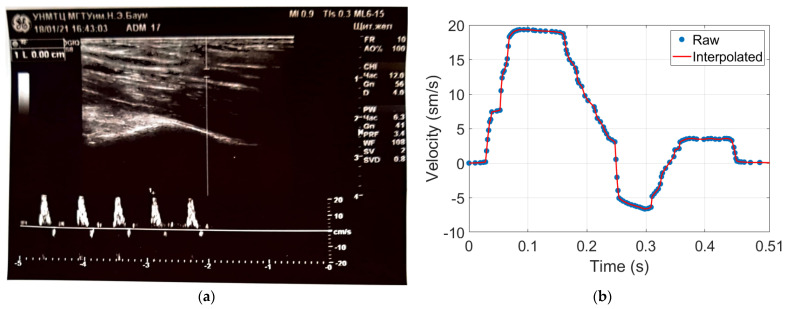
The obtained data using Doppler US: (**a**) brachial blood velocity assessment during 5 cardio cycles; (**b**) the averaged blood velocity curve.

**Figure 6 sensors-21-08438-f006:**
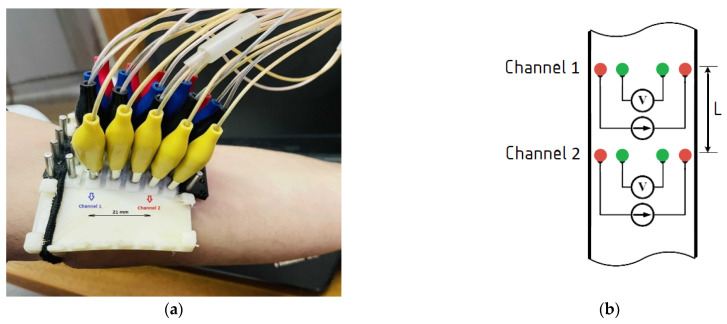
The electrode system prototype: (**a**) multichannel electrode system placement; (**b**) the layout scheme of the relative electrodes’ positions and channels in the region of interest.

**Figure 7 sensors-21-08438-f007:**
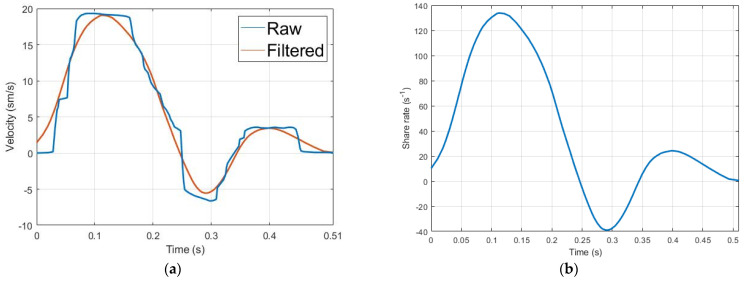
The red blood cells’ orientation effect estimation; (**a**) the averaged and smooth arterial blood velocity curve; (**b**) blood share rate curve.

**Figure 8 sensors-21-08438-f008:**
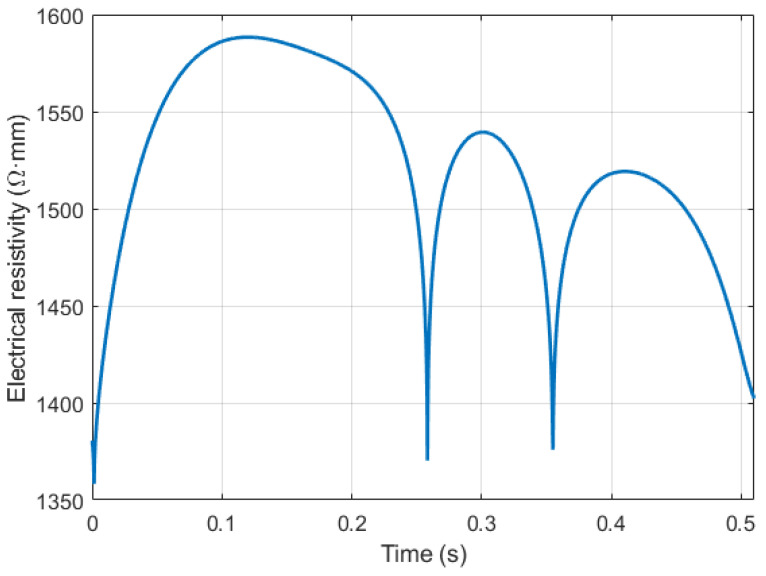
The blood resistivity change caused by red blood cells’ orientation effect.

**Figure 9 sensors-21-08438-f009:**
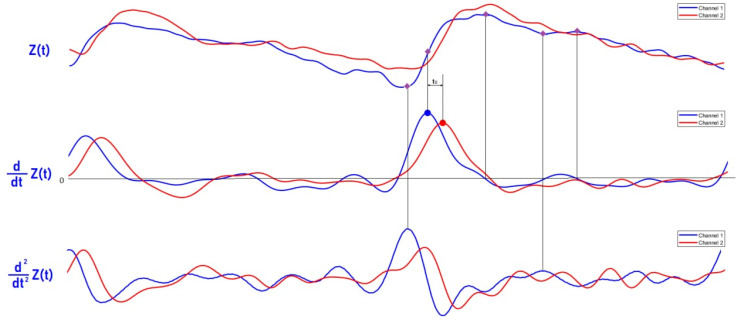
The recording of the bio-impedance signal with the first and second derivative.

**Figure 10 sensors-21-08438-f010:**
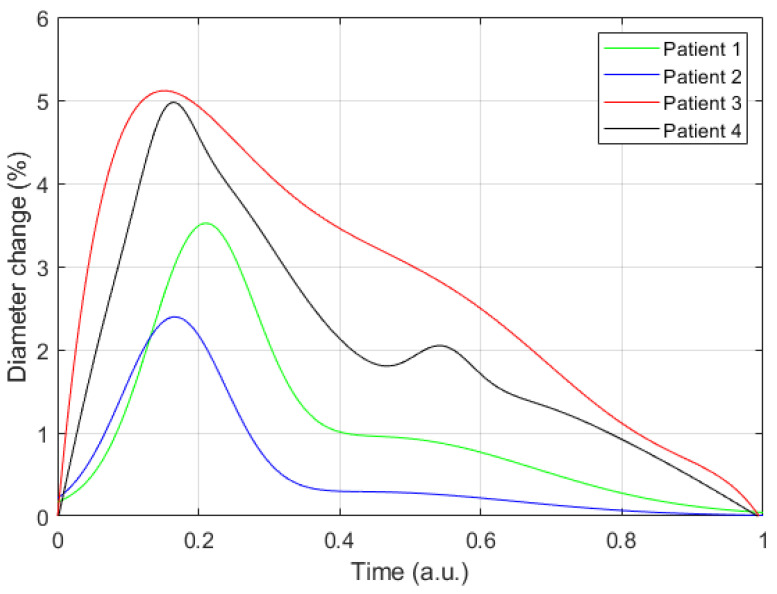
The resulting curves of diameter changing for four patients.

**Table 1 sensors-21-08438-t001:** Blood pressure estimation techniques.

Method	Features	Limitation
Auscultation	-It is possible to take into account individual physiological characteristics of the body [18]. -Low measurement errors during patient movements [19]. -Devices often do not require a power supply [20]. -Confirmed correlation with invasive method [21].	-Sensitive to indoor noise, to the friction of the cuff on clothes, to the location of a microphone [22]. -Require high qualifications [20]. -The unacceptability of the method for 5–10% of patients with deaf tones [19]. -Inaccuracy of measurements with low stiffness of artery walls [23]. -Not suitable for long-term monitoring. -Difficult to automate [22]. -Time consuming [10]. -Correct results depend on cuff size [21].
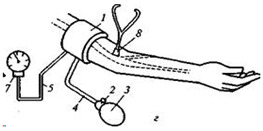		
Oscillometric	-Is recommended for clinical use by WHO [20]. -It is possible to take measurements under a small layer of clothing and measure pressure for patients with weak and infinite tones [19]. -Placement is not critical [9]. -External noise does not affect results [9]. -Automatic pressure estimation [22].	-Sensitive to mechanical vibrations, hand movements, to patient specificity [9,24]. -Large errors if patient has cardiac rhyme disease [24,25]. -Various results with different devices [9]. -Accuracy depends on algorithm [26]. -Importance of training personnel [20]
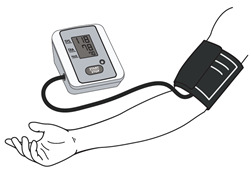		
Palpatory	-Devices often do not require a power supply [20]. -Suitable for noisy environment measurements [27].	-Sensitive to tremors, severe obesity, shivering [27]. -Not suitable for long-term monitoring. -Issues with diastolic arterial pressure assessment [27]. -Frequent measurements can harm the vessels. -Not suitable for diastolic pressure estimation [27].
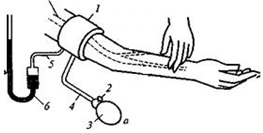		
Compensatory	-Non-invasive [28]. -Suitable for long-term monitoring [9]. -Small size of device and cuff. -Accurate measurements [9]. -Attaching to a finger.	-High cost [9]. -Sensitive to limb temperature. -Calibration is required [28]. -Artefacts are possible during measurement. -Not suitable for deep-seated arteries measurements [29].
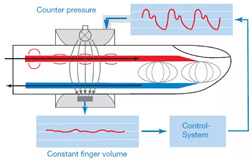		
Tonometry	-The pressure sensors are pressed directly against the skin [30]. -Non-invasive [30]. -Does not stop blood flow [31]. -Less sensitive than finger cuffs to vasoconstriction and vascular disease [10].	-High cost [30]. -Sensitive to device position, outdoor noise [30]. -Calibration required [9]. -Limited area of measurement due to instrument size [30]. -Not suitable for long-term and deep-seated arteries measurements [9]. -Not suitable for arteries without supportive bony structures [14].
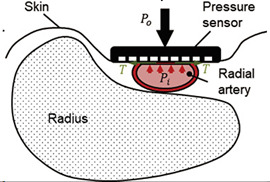		
Ultrasound	-Non-invasive [32]. -Possibility of deep tissue observation [14]. -The possibility of creating portable devices [14]. -The ability to determine the speed of blood flow [33]. -A high degree of research of the method [32].	-Requires ultrasound [32]. -Issues with diastolic arterial pressure assessment. -Sensitive to transducer location [33]. -The bulkiness and rigidity of ultrasound probes [14]. -Korotkov’s tones must be determined. -The occurrence of interference with minor movements [9]. -The difficulty of positioning [32].
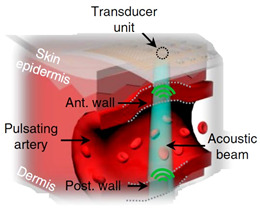		
Pulse wave velocity	-The simplicity of carrying out measurements. -Cuffless method [34]. -Non-invasive [34]. -Suitable for long-term monitoring [34].	-Confirmation of correlation between pressure and pulse wave velocity depends on model [34].
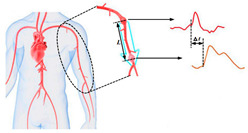		

**Table 2 sensors-21-08438-t002:** The technical specification of Reo-32.

Parameter	Value
Number of impedance measurement channels	31
Number of ECG channels	1
Sample frequency	500 Hz
Measuring scheme	Tetrapolar
Current amplitude	3 mA
Current frequency	100 kHz
The pulse measured range	2 Ohm
The base measured range	1–250 Ohm
The pulse impedance measuring accuracy	1 mOhm
The base impedance measuring accuracy	50 mOhm
The ECG measuring accuracy	3 uV

**Table 3 sensors-21-08438-t003:** The obtained results from experimental studies.

D_D_, mm	Depth, mm	$\rho_{1},\;\mathrm{Ohm}\cdot m$	$\Delta \rho_{2},\;\%$	E, kPa	dZ, Ohm	ΔD, %	D_S_, mm
3.8	8.0	3.8	14.9	78.6	0.13	4.9	4.0
3.9	5.5	5.6	15.8	94.0	0.45	3.4	4.1
3.3	6.5	4.5	17.1	122.2	0.12	2.4	3.4
3.1	3.1	2.9	16.8	84.9	0.29	5.1	3.3

## Data Availability

The data presented in this study are available on request from the corresponding author.

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
