# Peer review of "Bio-Impedance Sensor for Real-Time Artery Diameter Waveform Assessment"

_sensors, 2021, doi:10.3390/s21248438_

Round 1
Reviewer 1 Report
This paper aims to present a new device for non-invasive artery diameter waveform evaluation.
The paper suffers from some supplemental Major Issues that should be discussed before considering it for publication in Sensors.
MAJOR ISSUES:
- The topic of the paper could be of interest. However, the manuscript simply describes an experiment enrolling a single subject. Accordingly, the work seems to be too preliminary to be published as an original article. Authors are encouraged to expand their work by enrolling more subjects.
- Another great limitation of the work is the lack of validation. Authors should consider acquiring the true diameter waveform (by ultrasound imaging or other medical imaging techniques) and the true pressure waveform (by tonometry or other invasive/non-invasive approaches) in order to compare them with the waveforms obtained by the proposed system.
- The mathematical model was described in very general terms. Accordingly, it is not completely clear how the diameter waveform was obtained starting from bio-impedance signal, with or without the external data obtained with an ultrasound device.
- Results about the estimation of effects of red blood cell orientation were not reported.
Author Response
Dear reviewer
Thank you very much for your attention and time, please see the attachement
with respect
Dr.Mugeb

Reviewer 2 Report
Very interresting idea and bioimpedance application, that might be very important for rapid pressure changes monitoring. The idea of estimation of artery blood flow by means of bio-impedance is in my opinion very attractive and might be practically applied.
However In the paper several aspects should be considered / improved.
First of all - I am not native English but I can see some typos that should be corrected. (e.g. line 42 "high correlated", line 172 - unclear statement / gramma ? ,
Second , there is a lack of explanation of variables used in the equations - e.g. what are a,b in eq 1 and 2 ?? Please improve.
Next problem - Authors propose the use of bioimpedance. Provided model assumes point electrodes, where in practical implementation electrodes have certain areas. Higher areas - lesser current density under electrode but non-zero area electrode introduces spatial integration - altering so-called sensitivity function. Additionally - a electrode impedance itself should be considered.
Authors assume goal to produce long term wearable device - so electrode area, shape and location is important.
Next - the location of sensor - apparently in the area of interest there is a lot of conductivity distribution changes caused by the muscles and tendons. Thus mobile application is limited in such case - or provided algorithms will be able to separate such information.
Next - authors performed primary experiments - they used measurement system able to apply 3mA current at 100kHz. In my opinion 3mA is quite high current - especially for long-term wearable. Additionally frequency used - 100kHz is over so called beta-disspersion where cell shells brings no information. I do suggest try to use lower frequency (20-50kHz) if applicable.
The use of (a.u.) for time in fig. 5b should be commented - it is unclear how to relate it to another results.
In the further study author should also consider so called sensitivity function. In such electrode configuration it is possible to obtain positive and negative sensitivity depending on the electrode vs vessel location.
Author Response
Dear Reviewer
Thank you very much for your attention and your time, please see the attaachement
with respect
Dr.Mugeb

Reviewer 3 Report
The article with the title ‘Bio-impedance Sensor for Real-Time Artery diameter waveform Assessment’ presents the development of a self-calibrated bio-impedance-based sensor, which can provide a regular measurement of the blood pressure dependence time variable parameters such as the artery diameter waveform and the elasticity. Moreover, It is appreciable that the authors have also added the mathematical model of their proposed design. However, the following comments will add value to the manuscript.
- The multichannel electrode system’s measurement principle should also be discussed.
- Savitzky-Golay filter is used to filter the data in figure 5a. Why specifically this filter was used?
- Why is there a significant difference in channel 1 and 2 measurements of impedance in figure 6a?
- A short video for this experiment can add value to the manuscript and further make it easier for the readers to understand it.
- It is hard to understand the exact contribution of this paper? Mathematical Model? the proof of concept? Or both?
- How this work is different from [23-25] cited in the manuscript?
- A comparison table of the carried out work with reported papers should also be given.
- There are various grammatical mistakes, which should be corrected in the revised version.
Author Response
Dear reviewer
Thank you very much , please see the attachment
with respect
Dr.Mugeb

Round 2
Reviewer 1 Report
I appreciate some changes proposed by authors according to the issues raised with the first review. In particular, methods about mathematical models and results of red blood cells orientation are more clear. However, two very important issues remained unsolved. Firstly, the sample size has been increased but it is still too limited to validate a new system. Furthermore, the main limitation of the paper was not addressed as the system was not compared/validated against a reliable gold standard.
Author Response
Dear Reviewer
Thank you very much for your attention and your time . I will carefully consider all your comments in the next step of this work
With respect
Dr.Mugeb
Reviewer 3 Report
The revised version qualifies for the publication in Sensors Journal of MDPI.
Author Response
Dear Reviewer
Thank you very much for your full attention and your time . I appreciate your comments and will take your suggestions into consideration in the next step of this work
With respect
Dr.Mugeb